# SINGLE-CELL CAPSULE ATTENTION : AN INTERPRETABLE METHOD OF CELL TYPE CLASSIFICATION FOR SINGLE-CELL RNA-SEQUENCING DATA

## ABSTRACT

Single-cell RNA-sequencing technique can obtain genes' expression level of every cell. Cell type classification (also known as cell type annotation) on single-cell RNA-seq data helps to explore cellular heterogeneity and diversity. Previous methods for cell type classification are either based on statistical hypotheses of gene expression or deep neural networks. However, the hypotheses may not reflect the true expression level. Deep neural networks lack interpretation for the result. Here we present an interpretable neural-network based method single-cell capsule attention(scCA) which assigns cells to different cell types based on their different feature patterns. In our model, we first generate capsules which extract different features of the cells. Then we obtain compound features which combine a set of features' information through a LSTM model. In the end, we train attention weights and apply them to the compound features. scCA provides a strong interpretation for cell type classification result. Cells from the same cell type share a similar pattern of capsules' relationship and similar distribution of attention weights for compound features. Compared with previous methods for cell type classification on nine datasets, scCA shows high accuracy on all datasets with robustness and reliable interpretation.

## 1 INTRODUCTION

Genes' expression on celluar level provides lots of information for us to explore homogeneity and heterogeneity among different cells(Liang et al., 2014; Muraro et al., 2016; Baron et al., 2016). Traditional bulk sequencing technique can only measure the average gene expression level of all cells in a sample. Compared with bulk data, single-cell RNA-sequencing(single-cell RNA-seq) technique gives a more accurate measurement of every cell. Though single-cell RNA-sequencing is a rising technique, it is not mature enough and still has some limitations. For example, single-cell RNA-seq data has high 'dropout'. 'Dropout' means zero or low read counts in the data because many genes' expression are hard to detect(Huang et al., 2018; Pierson & Yau, 2015). Also, single-cell RNA-seq data are of high dimension because thousands of genes are in the transcriptome. These all bring challenges for studies on single-cell RNA-seq data. So far, many data mining and analysis methods have been applied on single-cell RNA-seq data to solve these challenges and explore more information on cells and genes(Svensson et al., 2018; Han et al., 2018; Lopez et al., 2018; Hwang et al., 2018; Guo et al., 2015).

Cell type classification is one of the most important tasks in single-cell RNA-seq data analysis. It helps to identify different cell types and explore the cellular heterogeneity. Previous methods on cell type classification can be categorized into methods based on statistical distribution hypotheses of gene expression and deep learning methods based on neural networks(Abdelaal et al., 2019). These methods lay a solid foundation for cell type classification. However, they still have limitations and weaknesses. Methods based on statistical hypotheses rely on prior knowledge of marker genes or gene expression's distribution assumptions. Up to now, there is no widely recognized hypotheses of gene expression. Deep neural network(DNN), a fast and effective learning method, is widely used in various fields. DNN does not rely on distribution hypotheses, however, it's hard to provide an reasonable explanation for the result behind the deep 'black-box'(Lin et al., 2017). ACTINN(Ma & Pellegrini, 2020), a deep neural network based method for cell type classification, uses three hidden

layers in its architecture. Though its model can predict cell types, the model still lacks reliable interpretations for the result. The challenge is that DNN does not provide a recognizable pattern that is closely related to different cell types.

Here, we introduce a new neural network based method scCA for cell type classification. scCA provides an reasonable interpretation for its classification result. Our model learns features from cells and provides patterns behind these features that are related to cell types. The first part is feature extraction. Given the gene expression level in every cell, We use capsules to extract different features. A capsule is a vector that represents a feature, which captures a certain type of information of the cell. Capsules also serve as dimension reduction for single cell RNA-seq data. We calculate the Pearson coefficients between every two capsules and generate the coefficient heatmap of every cell. We discover that cells from same cell type share a similar pattern in the heatmap while cells from different cell types have different patterns. These capsules capture features that are important for cell type classification.

And then, we use a sequence to sequence model, bidirectional-LSTM, to generate compound features. Each compound feature combines a set of features' information. Furthermore, we train multiple attention weights for these compound features. Then we multiply these weights to the compound features and generate the classification result. Attention weights help to lay more emphasis on important compound features. The larger the attention weight is, the more crucial its corresponding compound feature is. we also draw heatmaps for every cells' attention weights. We discover that cells from same cell type have similar patterns in the heatmaps of attention weights. This leads to a better classification performance and also provides an reliable interpretation behind the network.

We evaluate scCA on several datasets and compare with previous work. Our method shows high accuracy and robustness on all datasets.

The main contributions of our article are as follow: **We propose an interpretable neural network based method scCA for cell type classification. Through heatmaps of capsules' Pearson coefficients and attention weights, we find that cells from same cell type share a similar pattern. scCA achieves high accuracy and stable performances on all datasets.**

## 2 RELATED WORK

**Statistical Hypotheses Based Model**

Many machine learning methods with statistical hypotheses of gene expression level have been proposed for cell type classification. Some of these methods suggest distribution hypotheses of gene expression level and employ machine learning models based on these hypotheses. Some build a hierarchy with prior knowledge and statistical hypotheses. Moana takes advantage of KNN and SVM model(Wagner & Yanai, 2018). It first preprocesses data with KNN, then it uses dimension reduction method to extract useful features, at last it uses a Support Vector Machine classifier. In addition, Moana uses marker gene as a prior knowledge for cell type identification. Different from Moana, Garnett proposes a model based on tree hierarchy. It discovers the relation between cell types and subtypes with a tree structure(Pliner et al., 2019). With the help of marker gene, it builds a linear model to classify cell types. CellAssign builds a probabilistic model based on prior knowledge to classify cell types(Zhang et al., 2019b). SCINA makes use of EM algorithm and the information of marker gene to accomplish clustering as well as cell type identification task(Zhang et al., 2019a). scPred extracts features using SVD, then with a nonlinear kernel SVM, it predicts different cell types(Alquicira-Hernandez et al., 2019). However, these methods have their limitations. Their hypotheses such as ZINB(Zero-inflated Negative Binomial) or NB distribution of gene expression are not the exact reflections of gene 's expression levels in cells. Actually, there is not a consensus statistical distribution for single-cell RNA-seq data. Also they need prior knowledge like marker gene for cell type classification.

**Neural Network Based Models**

Deep neural network(DNN) based methods have been applied to cell type classification task. They are fast and robust. Neural networks don't heavily rely on prior knowledge and statistical hypotheses. ACTINN uses three fully connected layers and ReLU, softmax activation function to classify different cell types(Ma & Pellegrini, 2020). scCapsNet builds a capsule network to extract features

from the data and use dynamic routings among capsules to get a classification result(Wang et al., 2020). However, neural networks lack interpretation for their result. Through training the whole network by propagation, it's hard to explain every component's contribution for the result.

## 3 PRELIMINARIES

**Capsule Network and Dynamic Routing**

Hinton, Sabour and other co-authors first proposed Capsule network and dynamic routing in computer vision(Sabour et al., 2017). A capsule is a set of neurons or a vector that can represent an entity. Dynamic routing is the mechanism between capsule's layers. As they point out, the capsules in the layer above are parents and the capsules in the layer below are children.(Hinton et al., 2000; 2011). Dynamic routing helps to send the results from capsules in children layer to its appropriate parents in the above layer. By adjusting the coupling coefficients between capsules in two layers, the network learns which parent capsule is more important for the children capsule. Here in scCA, we use capsules to capture different features of the cell. We use a vector for implementation of capsule in our model. However, different from dynamic routing, we use LSTM and attention as the architecture for cell type classification.

**Long Short-Term Memory and Attention**

Long Short-Term Memory(LSTM) is a model widely used in Natural Language Processing(NLP)(Hochreiter & Schmidhuber, 1997). It is a sequence to sequence model. The inputs are usually the word embeddings(representations) of a sentence and the output is a sequence of representations. Each output representation not only contains information from its corresponding input, but also information from representations in the context, especially its neighbor inputs.

In our method, after extracting different features into capsules, we input these capsules into the LSTM model and obtain the output sequence. Every vector in the output sequence contains information of its corresponding input feature as well as other features. We name these vectors **compound features**. Some of the compound features have closer relationships with certain cell types. Inspired by Attention(Vaswani et al., 2017) mechanism, we apply attention weights to compound features in order to differentiate and emphasize those important compound features for classification.

## 4 SINGLE-CELL CAPSULE ATTENTION

In this section, we provide a detailed description of single-cell Capsule Attention(scCA). We will also give a vivid and transparent explanation for our classification result. scCA can be divided into three parts. The figure below shows its architecture.

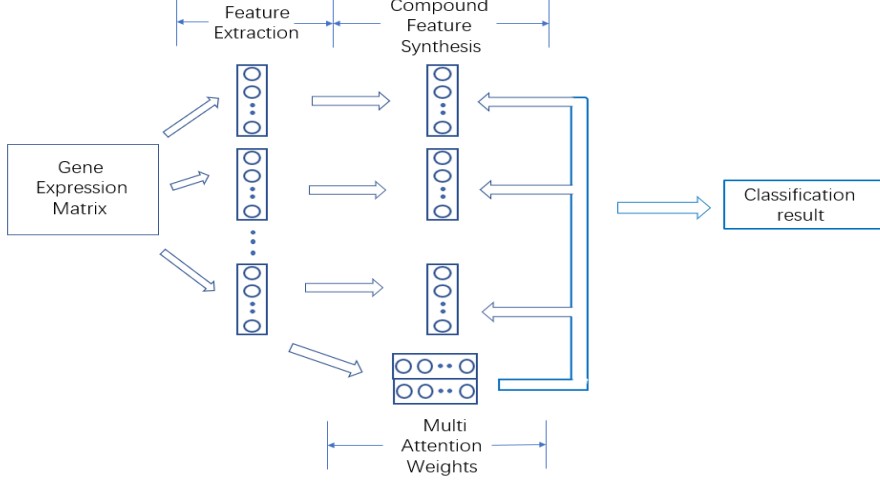

Figure 1: scCA framework

## 4.1 FEATURE EXTRACTION

The first part of scCA is feature extraction. Given thousands of genes' expression levels for a cell, we hope to learn simple but efficacious features for classification. Capsules not only serve as dimension reduction, but also extract a type of information of the cell. Through fully connected layers, we generate capsules to extract different features.

$$capsule_i = ReLU(W^i x + b^i), i \in [1, 2 \cdots m] \tag{1}$$

In this formula, $m$ represents the number of capsules. $W$ and $b$ are the weight matrices of fully connected layers. $ReLU$ is the activation function we use.

Every capsule contains a feature of a cell. We hope to discover different patterns of these features for different cell types. We calculate the Pearson coefficients between every two capsules. The Pearson coefficient is an indication of relationship between two features behind capsules. For every cell, we come to a matrix where every element is a Pearson coefficient of two capsules. We then draw a heatmap of the matrix. We discover that cells from same cell type share a similar pattern while cells from different cell types differ from each other.

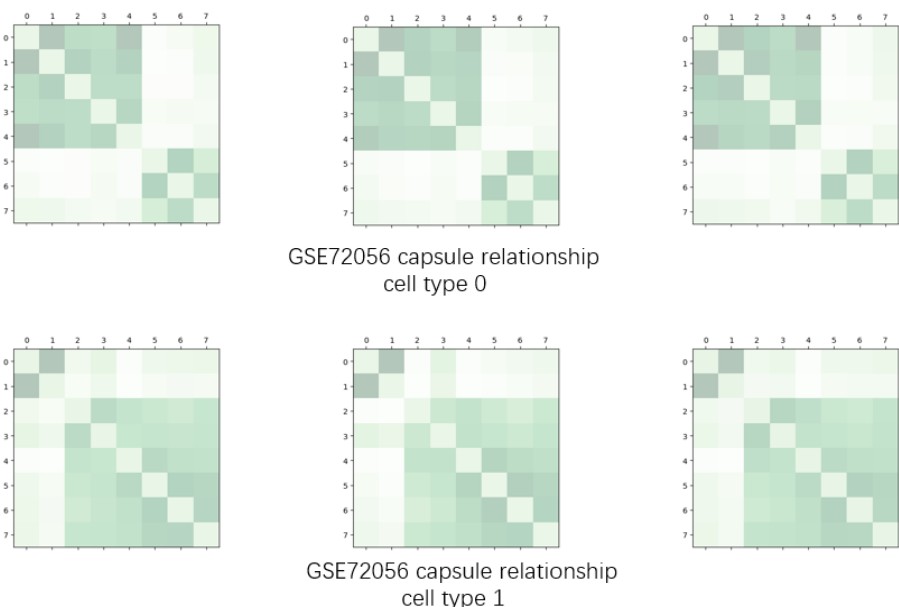

Figure 2: GSE72056 capsule relationship heatmap

Taking six cells from dataset GSE72056 as examples, we can see cells from same cell type have a similar pattern of capsules' relationship. The deeper the color is in the heatmap, the closer the relationship of two capsules is. The pattern is distinguishable among different cell types. From the figures above, we conclude that there exists some particular feature patterns for different cell types. Our feature extraction really extracts useful and discriminative features.

## 4.2 LSTM FOR COMPOUND FEATURES

Single feature is not enough for cell type classification because cells from different cell types may share same features. It's the specific combination of features that decides the cell type. Moreover, some features may have special relationships with other features. We need to explore the special patterns behind the features that determine cell types. The motivation for compound features lays in here. A LSTM model captures other features' information in the context. The capsules generated in feature extraction serve as the input of LSTM. For every feature in the sequence, a LSTM model takes in hidden information from other features as well as its own information. Then LSTM model

outputs a sequence of compound features. For implementation, we use a bi-directional LSTM model for training which is more stable and efficient.

$$[compound\_feature1, compound\_feature2, ...] = LSTM[capsule1, capsule2, ...] \quad (2)$$

### 4.3 ATTENTION MECHANISM

In the last part of scCA, scCA learns attention weights for these compound features. We have two intentions for these attention weights. The first intention is that we want to eliminate the effect of capsule's order in the input sequence. LSTM model is widely used in natural language processing where input data is a sequence. However, for cell type classification, capsules or features' order in the sequence shouldn't influence the result. So we add attention to emphasize the compound features' information itself, rather than its position information in the sequence. The second intention is that some compound features may be more significant in cell type classification. The attention weights for these compound features will be large, showing their importance. In scCA, we implement multi-attention and apply them to these compound features. We generate several combinations of compound features based on different attention weights. Then we use the norms of these combinations to serve as the final classification result.

$$sum\_of\_feature = sum(capsule_i), i \in [1, 2 \cdots m] \quad (3)$$

$$attention\_weight_i = ReLU(W\_attention^i * sum\_of\_feature + b\_attention^i), i \in [1, 2 \cdots cell\_type\_num] \quad (4)$$

$$classification\_result = L2\_norm(attention\_weights * compound\_feature) \quad (5)$$

For implementation, we first add all capsules together. Then we use fully connected layers to generate attention weights. In formula(3), *m* represents the number of capsules. In formula(4), *cell_type_num* represents the number of cell types in the dataset. We multiply the compound features which are generated by bi-directional LSTM by the attention weights. Then we use the L2 norm to get a classification result. Adding attention weights in the model structure lays more emphasis on important compound features. To provide more illustration for this, We also draw heatmaps of attention weight matrix for every cell. We find that cells from same cell type share a similar pattern of heatmap, which means certain cell type has their particular attention combinations of compound features.

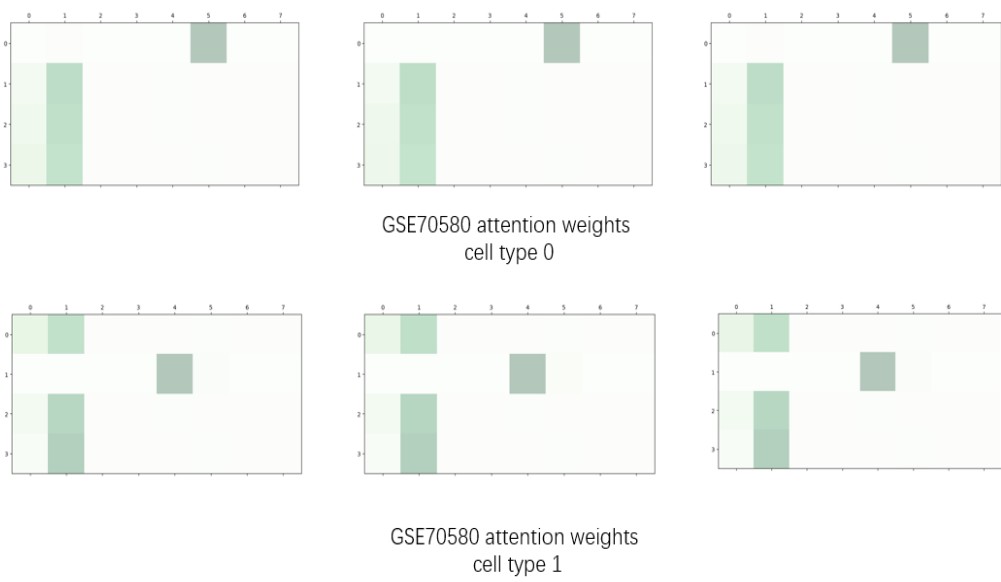

GSE70580 attention weights
cell type 0

GSE70580 attention weights
cell type 1

Figure 3: GSE70580 attention weights heatmap

The figure above shows 6 cells from dataset GSE70580. Three cells come from cell type 0 while the other come from cell type 1. The deeper the color is in the figure, the larger the attention weight is. We can find that attention weights' distribution vary from one cell type to another. Different cell types have different attention weight combinations of compound features. This provide reliable interpretation for our model.

## 5 EXPERIMENTS

### 5.1 INTRODUCTION OF DATASETS

Every dataset for single-cell RNA-sequencing consists of a gene expression matrix and cell type labels. Gene expression matrix is the input of our model. Every line in the matrix shows different genes' expressions in a cell. The figure below demonstrates a sketch heatmap of gene expression matrix after data preprocessing mentioned in next section. The higher the expression level is, the deeper the blue color will be. There are many places in the matrix that are shallow, which are the reflection of dropouts.

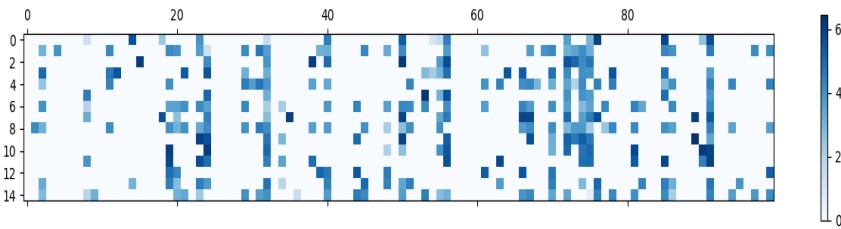

Figure 4: gene expression matrix heatmap

We then elaborate the nine datasets we use in experiments, listing out their numbers of cells and genes as well as the numbers of cell types.

Table 1: Single-Cell RNA-Seq data description

| Dataset | cell number | gene number | cell type number |
|---|---|---|---|
| CRC | 8496 | 12547 | 20 |
| GSE70580 | 647 | 26087 | 4 |
| GSE72056 | 4636 | 22280 | 7 |
| GSE75688 | 515 | 27420 | 5 |
| GSE96993 | 334 | 10827 | 4 |
| NSCLC | 9051 | 12415 | 16 |
| PBMC | 5356 | 14218 | 5 |
| Spleen human | 4406 | 14064 | 7 |
| Spleen mouse | 4432 | 12699 | 7 |

### 5.2 PREPROCESSING

There are tens of thousands of genes in the dataset. Before we train our model, we first implement data preprocessing. For every cell in the data, we first divide every gene's expression level by the sum of gene expression levels in the cell. This makes genes' expression levels among cells comparable. Then, we multiply the gene expression levels with a constant 100000 and employ a log transform to adjust the expression level in the data into a reasonable range.

## 5.3 PARAMETER SETTING

For feature extraction part, we generate 8 capsules through fully connected layers. Each of the capsule has a dimension of 128. We add L2 normalization to these capsule vectors and we use a single layer bidirectional LSTM model to generate compound features. We set the output dimension of LSTM to 16 and dropout of LSTM to 0.3. A bidirectional LSTM has two output sequences. We add them together and get a single sequence of compound features. To obtain attention weights, we first add all capsules together. Then we input the sum of capsules into fully connected layers. The output dimension of these layers is also the number of capsules. The number of attention weights is set to the same as cell type number in the dataset. We then multiply the attention weights to the compound features. After multiplying the attention weights to the compound features, we obtain a matrix whose row number equals to the number of cell types and column number equals to the dimension of compound features. Then we calculate the L2-norm of every row and apply Softmax function. In this way, we obtain a vector whose dimension equals to the number of cell types in the dataset and the sum of the elements in the vector is 1. This vector is the output of scCA. During the training process, cross entropy loss is used as the loss function. The activation function in our model is ReLU. We set the dropout to 0.3 and we also add Softmax to the classification result. We split the dataset into training set and test set with the ratio of 4:1. The model is trained for 50 epochs with a Pytorch implementation.

## 5.4 COMPARISON WITH OTHER METHODS

We first compare scCA with some machine learning methods, including Support Vector Machine(SVM), Linear Discriminant Analysis(LDA), Decision Tree(DT) and Random Forest(RF). From the table below, we can find that scCA has a robust performance with high accuracy on all datasets. It also has some outstanding performances on some of the datasets.

Table 2: Accuracy comparison with various machine learning methods

| dataset | scCA | SVM | LDA | DT | RF |
|---|---|---|---|---|---|
| CRC | 88.29% | 90% | 87.29% | 64% | 81.47% |
| GSE70580 | 96.92% | 97.69% | 96.15% | 90% | 96.92% |
| GSE72056 | 94.50% | 93.85% | 91.91% | 86.63% | 91.59% |
| GSE75688 | 92.23% | 92.23% | 91.26% | 88.34% | 91.26% |
| GSE96993 | 82.83% | 80.59% | 77.61% | 68.65% | 82.08% |
| NSCLC | 84.15% | 85.20% | 82.77% | 62.39% | 79.18% |
| PBMC | 98.32% | 98.13% | 97.48% | 95.52% | 97.94% |
| Spleen human | 92.85% | 92.85% | 87.86% | 82.53% | 87.86% |
| Spleen mouse | 96.84% | 97.29% | 94.58% | 88.61% | 92.33% |

Table 3: Accuracy comparison with various neural network methods

| Dataset | scCA | scCapsNet | fully connected network | denoising autoencoder |
|---|---|---|---|---|
| CRC | 88.29% | 84.47% | 88.64% | 83.71% |
| GSE70580 | 96.92% | 94.62% | 96.92% | 96.15% |
| GSE72056 | 94.50% | 92.20% | 93.53% | 91.33% |
| GSE75688 | 92.23% | 91.74% | 92.23% | 91.98% |
| GSE96993 | 82.83% | 77.61% | 80.59% | 79.59% |
| NSCLC | 84.15% | 79.18% | 82.93% | 79.80% |
| PBMC | 98.32% | 98.23% | 98.5% | 97.60% |
| Spleen human | 92.85% | 91.40% | 92.63% | 89.79% |
| Spleen mouse | 96.84% | 96.17% | 97.64% | 95.41% |

We then compare scCA with neural network based methods. Here we implement fully-connected network and denoising autoencoder. We use the code provided in scCapsNet. From the table above,

we can see that among models based on neural networks, scCA still has a good performance. It outperforms other methods on most of the datasets.

In conclusion, through the comparison of our model with other models, scCA shows robustness and high accuracy. On some datasets, scCA has the best performance of all models.

## 6 INFLUENCE OF DIFFERENT PARAMETERS

In our model, the number and dimension of capsules, the dimension of compound features can be adjusted. To test their influence on scCA's performance, we carry out experiments under different combinations of parameters.

Table 4: scCA's performance under different parameters with 128-dimension capsule

| Dataset | 8capsule + 16-dim compound features | 16capsule + 16-dim compound features | 8capsule + 32-dim compound features | 16capsule + 32-dim compound features |
|---|---|---|---|---|
| CRC | 88.29% | 88.41% | 88.58% | 88.94% |
| GSE70580 | 96.92% | 96.92% | 96.15% | 96.92% |
| GSE72056 | 94.50% | 93.96% | 94.07% | 94.18% |
| GSE75688 | 92.23% | 92.23% | 92.23% | 92.23% |
| GSE96993 | 82.83% | 82.08% | 83.58% | 82.78% |
| NSCLC | 84.15% | 84.15% | 83.76% | 84.70% |
| PBMC | 98.32% | 98.32% | 98.32% | 98.60% |
| Spleen human | 92.85% | 92.40% | 92.29% | 92.85% |
| Spleen mouse | 96.84% | 96.73% | 97.06% | 96.73% |

In table 4, we set the capsule dimension to 128 and change the number of capsules and the dimension of compound features. Through the result from the table, we discover that there is no visible improvements no matter we rise the number of capsule or the dimension of compound features.

Table 5: scCA's performance under different parameters with 64-dimension capsule

| Dataset | 8capsule + 16-dim compound features | 16capsule + 16-dim compound features | 8capsule + 32-dim compound features | 16capsule + 32-dim compound features |
|---|---|---|---|---|
| CRC | 87.70% | 88.17% | 88% | 88.29% |
| GSE70580 | 96.53% | 96.92% | 96.53% | 96.92% |
| GSE72056 | 93.64% | 94.07% | 93.75% | 94.07% |
| GSE75688 | 91.26% | 91.26% | 92.23% | 92.23% |
| GSE96993 | 82.08% | 82.83% | 81.34% | 83.58% |
| NSCLC | 82.21% | 82.44% | 82.93% | 84.32% |
| PBMC | 98.32% | 98.41% | 98.04% | 98.41% |
| Spleen human | 92.06% | 92.17% | 91.95% | 92.74% |
| Spleen mouse | 96.73% | 96.73% | 96.61% | 96.92% |

In table5, we set the capsule dimension to 64 and change the number of capsules and the dimension of compound features. Together with table 4, we discover that number of capsules, the dimension of capsules and dimension of compound features have little influence on scCA's performance. Our model performs well under different settings of parameters.

## 7 CONCLUSION

In this article, we provide a neural network based method scCA for single-cell RNA-seq data cell type classification. scCA uses capsules to extract features from cells. Then through a LSTM network it generates compound features which contain information from a set of single features. By applying attention weights to these compound features, we lay emphasis on compound features which are more closely related to cell types. Deep neural network based, scCA provides reliable interpretations for its result. We calculate the Pearson coefficients among capsules and draw heatmaps of these

coefficients. We discover that cells from same cell types share a similar pattern of capsules' relationship while cells from different cell type differ from each other. Moreover, we also draw heatmaps of the attention weights. We also find out that same cell types emphasize on same compound features. These provide vivid explanations and illustrations for our model. We test scCA on nine datasets. Compared with other machine learning and deep learning methods, scCA shows robustness and has a high classification accuracy. Furthermore, we adjust the parameters in our model and discover that our model is not sensitive to them.

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
