# OpenReview forum: "Single-Cell Capsule Attention : an interpretable method of cell type classification for single-cell RNA-sequencing data"
_ICLR.cc/2022/Conference — ICLR 2022 Submitted_

### Official Review · Reviewer_mrct · 2021-10-29

**Correctness:** 4
**Technical Novelty And Significance:** 2
**Empirical Novelty And Significance:** 1
**Recommendation:** 1
**Confidence:** 3

**Main Review:**

The paper suggests a neural network architecture (combining capsules, attention, and LSTM) to perform cell type classification for single-cell RNA-seq data. The authors claim that their method performs well and is interpretable -- more so than the competitors.

I found the paper difficult to understand, classification results not convincing, and interpretability claims greatly overstated. Unfortunately I believe this is a clear reject.

Disclaimer: I have heard about attention and capsules but do not know how they work. However, I think this paper can be assessed even if one treats the model as a black box. Also, I do work with single-cell RNA-seq data all the time.


MAJOR COMMENTS

* The authors claim that their method is "interpretable" (it's right in the title) but then does not demonstrate how it can be interpreted in practice. How is it more interpretable than a fully-connected network, a SVM, or a random forest (that have similar classification performance: Tables 1-2)? Unclear.

* The authors claim that their method performs well, but in fact it does not perform better than a fully-connected network (Table 3), which does not use either attention, or capsules, or LSTM.

* The paper is hard to follow, e.g. I could not fully understand what exactly is shown in Figures 2 and 3. Figures do not have meaningful captions, subplots and axes are not labeled, surrounding text is hard to read.

**Summary Of The Paper:**

The paper suggests a neural network architecture (combining capsules, attention, and LSTM) to perform cell type classification for single-cell RNA-seq data. The authors claim that their method performs well and is interpretable -- more so than the competitors.

**Summary Of The Review:**

I found the paper difficult to understand, classification results not convincing, and interpretability claims greatly overstated. Unfortunately I believe this is a clear reject.

Disclaimer: I have heard about attention and capsules but do not know how they work. However, I think this paper can be assessed even if one treats the model as a black box. Also, I do work with single-cell RNA-seq data all the time.

---

> ### Author Response · Authors · 2021-11-20
> **Replies on the review**
>
> Thanks for your review and suggestions. Here are the replies and explanations of the questions.
> 1. * review : "The authors claim that their method is "interpretable" (it's right in the title) but then does not demonstrate how it can be interpreted in practice. How is it more interpretable than a fully-connected network, a SVM, or a random forest (that have similar classification performance: Tables 1-2)? Unclear." and "However, I think this paper can be assessed even if one treats the model as a black box."
> * reply : Thanks for your comment. From the Visualization result, we discover that cells from different cell types have different patterns, including the Pearson Coefficients heatmap and attention weights heatmap. This helps to provide interpretations for scCA.  We do not think fully connected network provides simple and obvious interpretation for single-cell RNA-seq data cell type classification. Fully connected framework contains larger parameter matrices which add difficulties for simple and reliable interpretation. Also, fully connected network does not have a cell-type specific pattern in its network as scCA.
>
> 2. * review : "The authors claim that their method performs well, but in fact it does not perform better than a fully-connected network (Table 3), which does not use either attention, or capsules, or LSTM."
> * reply : Thanks for your comment. From our experimental result, scCA achieves the highest accuracy among all methods we use in experiment on some datasets such as GSE72056, GSE96993. On other datasets, scCA has comparable performances to other methods, showing strong robustness.
>
> 3. * review : "The paper is hard to follow, e.g. I could not fully understand what exactly is shown in Figures 2 and 3. Figures do not have meaningful captions, subplots and axes are not labeled, surrounding text is hard to read."
> * reply :  Thanks for the comment. We present the descriptions of Fig2 and Fig 3 in the article. As for Fig2, " We calculate the Pearson coefficients between every two capsules. The Pearson coefficient is an indication of relationship between two features behind capsules. For every cell, we come to a matrix where every element is a Pearson coefficient of two capsules. We then draw a heatmap of the matrix. We discover that cells from same cell type share a similar pattern while cells from different cell types differ from each other." and "Taking six cells from dataset GSE72056 as examples, we can see cells from same cell type have a similar pattern of capsules' relationship. The deeper the color is in the heatmap, the closer the relationship of two capsules is. ".
>
> * For Fig3, "To provide more illustration for this, We also draw heatmaps of attention weight matrix for every cell. We find that cells from same cell type share a similar pattern of heatmap, which means certain cell type has their particular attention combinations of compound features." and "The figure above shows 6 cells from dataset GSE70580. Three cells come from cell type 0 while the other come from cell type 1. The deeper the color is in the figure, the larger the attention weight is. We can find that attention weights' distribution vary from one cell type to another."
>
> * The dimensionalities of the vectors are mentioned in 'Parameter Setting' part. We split the dataset into training set and test set with the ratio of 4:1.

---

### Official Review · Reviewer_YcRA · 2021-10-30

**Correctness:** 2
**Technical Novelty And Significance:** 1
**Empirical Novelty And Significance:** 1
**Recommendation:** 1
**Confidence:** 4

**Main Review:**

This work is a straightforward application of well-known methods to the single-cell RNA seq data and do not provide technical contributions

Results provide weak support to authors’ claims:
* The classification results show minor improvement compared to standard methods like SVM and fully connected network. No other insights are provided to support the benefit of using scCA to classify scRNA data.
* On the interpretability of scCA, only heat maps of feature correlations for two cell types are shown. First, It is not clear such visualization is unique to the capsule network in any way. I believe other methods can provide similar results, for example, by looking at the hidden layers of a fully connected network. Second, it is not clear such interpretable patterns can generalize to other cell types/datasets. Since the names of cell types are not specified, it is also difficult to judge the resolution of learned features.

The presentation of the paper needs to be improved. Particularly

* Missing important details. For example, it is not clear Fig 2 and 3 are generated. No train/test split/cross validation and parameter tuning are described.


**Summary Of The Paper:**

In this work, the authors implemented a capsule network to classify cell types for single-cell RNA-seq data. The capsule network was compared with standard methods such as SVM, LDA, and a fully connected neural network. The authors claim the capsule network provides interpretable results on cell type classification.

**Summary Of The Review:**

I think this work is a straightforward application of well-known methods to the single-cell RNA seq data. The results do not support the proposed method is superior to alternatives. I am therefore inclined to reject this paper.

---

> ### Author Response · Authors · 2021-11-20
> **Replies on the review**
>
> Thanks for your review and suggestions. Here are the replies and explanations of the questions.
> 1. * review : "The classification results show minor improvement compared to standard methods like SVM and fully connected network. No other insights are provided to support the benefit of using scCA to classify scRNA data."
> * reply : Thanks for your comment. From our experimental result, scCA achieves the highest accuracy among all methods we use in experiment on some datasets such as GSE72056, GSE96993. On other datasets, scCA has comparable performances to other methods, showing strong robustness.
>
> 2. * review : "On the interpretability of scCA, only heat maps of feature correlations for two cell types are shown. First, It is not clear such visualization is unique to the capsule network in any way..."
> * reply : Thanks for your comment. We do not think fully connected network provides simple and obvious interpretation for single-cell RNA-seq data cell type classification. Fully connected framework contains larger parameter matrices which add difficulties for simple and reliable interpretation. Also, fully connected network does not have a cell-type specific pattern in its network as scCA.
>
> 3. * review : "Missing important details. For example, it is not clear Fig 2 and 3 are generated. No train/test split/cross validation and parameter tuning are described."
> * reply : Thanks for the comment. We present the descriptions of Fig2 and Fig 3 in the article. As for Fig2, " We calculate the Pearson coefficients between every two capsules. The Pearson coefficient is an indication of relationship between two features behind capsules. For every cell, we come to a matrix where every element is a Pearson coefficient of two capsules. We then draw a heatmap of the matrix. We discover that cells from same cell type share a similar pattern while cells from different cell types differ from each other." and "Taking six cells from dataset GSE72056 as examples, we can see cells from same cell type have a similar pattern of capsules' relationship. The deeper the color is in the heatmap, the closer the relationship of two capsules is. ".
>
> * For Fig3, "To provide more illustration for this, We also draw heatmaps of attention weight matrix for every cell. We find that cells from same cell type share a similar pattern of heatmap, which means certain cell type has their particular attention combinations of compound features." and "The figure above shows 6 cells from dataset GSE70580. Three cells come from cell type 0 while the other come from cell type 1. The deeper the color is in the figure, the larger the attention weight is. We can find that attention weights' distribution vary from one cell type to another."
>
> * The dimensionalities of the vectors are mentioned in 'Parameter Setting' part. We split the dataset into training set and test set with the ratio of 4:1.

---

### Official Review · Reviewer_khq7 · 2021-11-01

**Correctness:** 3
**Technical Novelty And Significance:** 2
**Empirical Novelty And Significance:** 2
**Recommendation:** 3
**Confidence:** 4

**Main Review:**

The paper is well structured and relatively easy to follow and understand (at least on a high level).
However, I am having major concerns with several of the tenets of this work.

First of all I am not a specialist of capsule networks, and I may very well be missing something.
However, given that there is only one layer of capsules (that are then combined using an LSTM), there is no dynamic routing, which I understand are (at least partly) the essence of Capsule Nets.
As such, I am not convinced the suggested architecture really falls into the Capsule framework.
At the very least, it can be seen as a multi-headed single layer MLP that extracts features, which can be interesting.

I understand that the features extracted, or capsules need to be combined.
However, I do not believe that an LSTM is the right tool for that, given that they do not constitute a sequence.
The authors mention that this is an issue and suggest that they used attention to make the model invariant to the order of the capsules but I don't understand why that would be the case.
Can you elaborate?
In any case, along the lines of what the authors have in mind, I wonder if using Deep Sets (Zaheer et al. '18), which can be combined with attenion like in (Ilse et al. '18) to provide interpretability, is not a better candidate.
In all honestly, deep sets are mostly useful when the number of elements in the set is not constant across example.
Here, I suspect even a concatenation of the capsules with a MLP should do the trick.

An important claim of the paper is that the suggested model offers interpretability.
However, I am not convinced at all that is the case.
The capsules are obtained from a non-linear combination of all the genes.
As such, I am not sure what benefit there is from observing that a given cell type is associated to a specific capsule pattern as those will have no clear interpretability.
I am maybe missing something, but in any case, despite the claims, the authors never comment on the interpretation of those features and whether they are biologically relevant or interesting.

It is interesting that some patterns are emerging from the capsule activations.
But in fact, the model is trained on a classification task.
As such, it is only natural that the features extracted will be made separable.
I don't think there is anything that is emerging specifically with the given architecture.
With a vanilla MLP, the features extracted in the late layers of the model would also exhibit such patterns because they would eventually need to be separable to be classified.

The presented experimental results are reasonable but certainly not groundbreaking, especially as they are mostly compared to vanilla methods such as SVM, Decision Trees, Random Forests etc.
They seem to mostly be on-par with other methods.
This would be fine if the suggested approach was providing further benefits or a very novel approach.
Perhpas that could be the case if there were more analysis on interpretability and scCA was highlighting some biolgically interpretable and relevant features but in the current manuscript, that is not sufficient.


Beyond those major points, here are a few more remarks:
 - Given the similarities with scCaps, there definitely should be more emphasis on the comparison between the two approaches.
 - Through the manuscript, understand the "mathematical" details of the paper is made difficult because the authors never mention the dimensions of the different vectors and matrices they define. Can you correct that?
 - At the end of the Statistical Hypotheses Based Model subsection, the sentence "Their hypotheses of gene expression may not match with the true expression level." is not clear to me. Can you elaborate?
 - It is also mentioned in the "preprocessing" that the data is "amplified". What does that mean?
 - In the "parameter setting" section, there is mention of a normalization. Can you elaborate on which?
 - Also, the writing in that section is really not clear to me, for instance :"Then, we get the classification result". What is actually meant here?
 - It is never mentioned which loss was used to train the model. Is it the cross-entropy loss?
 - In several instances, it is written that "Hinton" proposed Capsule networks, or "pointed out" something. It would be more approriate to refer to Sabour and co-authors (or the authors of author specific papers) rather than "name dropping" Hinton.
 - There are a couple typos that I suspect are due to reducing to fit into the conference format (e.g. at the top of page 2, "scCA provides *a* reasonable", "in every cell, *we* use" etc...)

**Summary Of The Paper:**

The paper tackles the problem of classifying single-cell RNA-seq expression data according to their underlying cell types.
To do so, they suggest a deep neural network, inspired from Capsule Networks, and making use of attention to provide interpretable results.
The features extracted by the network are empirically shown to exhibit cell-type-specific patterns.
In terms of performance, the suggested method compares favourably or is on par with classic, standard ML approaches.
Finally, the authors show that the results seem to be relatively stable with respect to the choice of some architecture hyper-parameters.

**Summary Of The Review:**

In its current form, the proposed approach does not seem groundbreakingly novel, nor significantly improve over existing methods, nor offer extra benefits.
I think some of the design choices are debatable and I would love to hear what the authors have to say to the different points I raised.
Without further justification, though, I think the contributions are too minor to justify an acceptance at ICLR.

---

> ### Author Response · Authors · 2021-11-20
> **Replies on the review**
>
> Thanks for your review and suggestions. Here are the replies and explanations of the questions.
> 1. *  "First of all I am not a specialist of capsule networks.." and "Given the similarities with scCaps, there definitely should be more emphasis..."
> *  Thanks for your advice. Dynamic routings helps to build the relationships among parent layer's capsules and children layer's capsules. By dynamic adjusting the coupling coefficients between layers, capsule network finds out which capsules in the parent layer contribute more to capsules in the children layer. scCA uses capsules in the first step to serve as feature extraction. Then, scCA uses BiLSTM to combine information from other capsules to get compound features. After that, with the attention mechanism, scCA lays more emphasis on the important compound features for cell type classification.
>
> 2. *  "I understand that the features extracted..."
> *  We use a Bidirectional LSTM with attention as the main framework. The intention of LSTM and attention is that we hope to combine every feature’s information with the information of other features and lay more emphasis on those important ones. Furthermore, we are considering of an appropriate self-attention mechanism for single-cell RNA-seq cell type classification. The preliminary results are satisfying.
>
> 3. * "An important claim of the paper is that the suggested model offers interpretability..." and "It is interesting that some patterns..."
> *  Thanks for your comment. As for model's interpretation, from the Visualization result, we discover that cells from different cell types have different patterns, including the Pearson Coefficients heatmap and attention weights heatmap. This helps to provide interpretations for scCA. Also, We do not think fully connected network provides simple and obvious interpretation for single-cell RNA-seq data cell type classification. Fully connected framework contains larger parameter matrices which add difficulties for simple and reliable interpretation. Fully connected network does not have a cell-type specific pattern in its network as scCA.
>
> 4. *  "The presented experimental results are reasonable but certainly not groundbreaking,..."
> * From our experimental result, scCA achieves the highest accuracy among all methods we use in experiment on some datasets such as GSE72056, GSE96993. On other datasets, scCA has comparable performances to other methods, showing strong robustness. However, we will try to explore more on biological interpretations in future work. Thanks for the helpful suggestions.
>
> 5. * "Through the manuscript, understand the "mathematical" details of the paper is made difficult.."
> * Thanks for the comment. We mention the dimensions of vectors and matrices in the "Parameter setting" section.
>
> 6. * "At the end of the Statistical Hypotheses Based Model subsection, the sentence "Their hypotheses..."
> * Some previous models assume that gene expression in cells subjects to ZINB (Zero-inflated Negative Binomial) or NB distribution. However, ZINB or NB is only a hypothesis of gene expression level, but not the exact reflection of gene 's expression level in cells. Actually, there is not a consensus statistical distribution for single-cell RNA-seq data.
>
> 7. * "It is also mentioned in the "preprocessing" that the data is "amplified"
> * Here we would like to specify how we preprocess single-cell RNA-seq data. For every cell in the data, we first divide every gene’s expression level by the sum of gene expression levels in the cell. This makes gene’s expression level among cells comparable. Then, we multiply the gene expression level with a constant 100000 and employ a log transform to adjust the expression level in the data into a reasonable range. We have also revised and add detailed interpretations into the article.
>
> 8. * "In the "parameter setting" section, there is mention of a normalization"
> * Here we implement L2 normalization on the capsule vectors generated by the fully connected layers. We have also revised and add detailed interpretations into the article。
>
> 9. * "Also, the writing in that section is really not clear to me..." and "It is never mentioned which loss..."
> * After multiplying the attention weights to the compound features, we obtain a matrix whose row number equals to the number of cell types and column number equals to the dimension of compound features. Then we calculate the L2-norm of every row and apply Softmax function. In this way, we obtain a vector whose dimension equals to the number of cell types in the dataset and the sum of the elements in the vector is 1. This vector is the output of scCA. During the training process, cross entropy loss is used as the loss function.
>
> 10. * "In several instances, it is written that "Hinton" proposed Capsule networks..."
> *  Thanks for your comment and advice. Sorry for the vague information. We have revised the reference to the detailed and appropriate form in the paper.

---

> > ### Comment · Reviewer_khq7 · 2021-11-22
> > **Thanks for the reply. My assessment of the paper remains unchanged.**
> >
> > Thanks to the authors for answering my questions and providing some clarification. However, my main concerns about this submission remain.
> >
> > "This helps to provide interpretations for scCA."
> > But how? How do you interpret the different features learnt in the capsules? How do you relate them to biologically meaningful quantities?
> >
> > "Fully connected network does not have a cell-type specific pattern in its network as scCA."
> > Are you sure about that? A FCN trained on a classification task is optimized to make the outputs of its last layer (before the classification layer) separable in the same sense that scCA does. As such, they will surely display some cell-type specific patterns because that is precisely what they are trained for. This is definitely my expectation, and some other reviewers have expressed the same. If that does not hold true for this dataset, or that the patterns are not as clear as with scCA, I think it will be something interesting to display, but the burden is on the authors to show that there is more structure or patterns learnt by the capsules compared to FCN.
> >
> > I feel that my main concerns, which were also raised by the other reviewers, remain mostly unanswered. To be fair, I think addressing them in depth would require much deeper re-work than a rebuttal phase can allow.
> > In any case, I am choosing to keep my assessment and rating of the paper unchanged.

---

> > > ### Author Response · Authors · 2021-11-22
> > > **Reply on the review**
> > >
> > > Thanks for your comments and advice. As you point out, it will be more reasonable if we relate scCA’s capsules or compound features to some biologically meaningful quantities, such as specific marker genes. We will make improvements in future work. Also, we should specify more about scCA’s cell-type specific patterns and compare them with related work or a simple FCN. At last, we would like to express again our gratitude to your comments and review.

---

### Official Review · Reviewer_HMhu · 2021-11-02

**Correctness:** 2
**Technical Novelty And Significance:** 1
**Empirical Novelty And Significance:** 2
**Recommendation:** 3
**Confidence:** 5

**Main Review:**

Strengths of the paper

* Benchmarking is undertaken across a large number of datasets. These datasets span different number of cells and cell types, diverse biological systems

* The robustness of scCA is well tested. The broad range of parameters tested give a degree of confidence in the results.


Weaknesses of the paper

* The presented approach is unfortunately of minimal practical utility. Single-cell datasets have severe batch effects with technical differences between technologies, tissue handling etc. Therefore, a classifier trained on one dataset is not guaranteed to generalize to other datasets even if the data is derived from the biological system. Therefore as highlighted by the authors, most of the cell-type classification methods make use of known markers to ensure generalization. Other methods (such as scVI) use a semi-supervised approach rather than a fully supervised approach. Finally, there have been recent developments which use xGBoost to ensure generalization to different datasets (https://www.biorxiv.org/content/10.1101/2021.05.20.445014v1.full).

* The authors claim scCA provides greater interpretation compared to existing approaches. I do not believe the capsule features and compound features necessarily add to the biological interpretation of the cell-type gene expression states. The additional interpretation as claimed by the authors is not sufficiently elaborated.

* One of the main claims of the paper is that capsules encode similar information for cells of the same celltype. This claim is demonstrated by examples of few cells in Fig. 2. Given the centrality of the claim, the authors should elaborate and provide additional metrics to prove. The authors should also specify how the model architecture ensures this consistency.

* LSTM as noted by the authors is used for data with sequences. So, it is not clear why LSTM architecture is the choice here given there is no sequence data - infact, the authors adjust the weights to account for the fact that single-cell data and capsules are not sequential in nature. Therefore, it is not clear why LSTM is the model choice compared to alternatives.

* A few other issues in terms of single-cell data processing:

	- Given that single-cell data follows negative binomial distribution, variance is a function of the mean and hence use of most variable genes would not correct for the mean. Dispersion estimates are typically used as a work-around (https://genomebiology.biomedcentral.com/articles/10.1186/s13059-019-1874-1)

	- The gene expression map in Fig. 4 is rather unrepresentative since the dynamic range of scRNA-seq in linear scale spans 2-3 orders of magnitude

	- Is the single-cell data normalized for differing molecule counts in each cell?



**Summary Of The Paper:**

This manuscript describes scCA (single-cell capsule attention) for cell-type classification using single-cell RNA-seq data. scCA usesa combination of capsule networks, LSTN and attentions to undertake cell-type classifiation. Capsules extract features from single-cell data, the combination of capsules called "compound features" through LSTM and attention weights are employed. scCA is compared against a host of other techniques and performs well across 9 diverse single-cell datasets. scCA results are also robust to various parameter choices across datasets.

**Summary Of The Review:**

The main reasons for the recommendation are as follows:
* The manuscript does not demonstrate generalization from one dataset to another which is the main practical application of cell-type classification using single-cell data.
* The claims of additional biological interpretation are not sufficiently elaborated
* Given the subject specific nature of the submission, it might not be of broad interest to the ICLR audience.

---

> ### Author Response · Authors · 2021-11-20
> **Replies on the reviews**
>
> Thanks for your review and suggestions.
> Here are the replies and explanations of the questions.
> 1. * review : "The presented approach is unfortunately of minimal practical utility. Single-cell datasets have severe batch effects with technical differences between technologies, tissue handling etc.  ... Other methods (such as scVI) use a semi-supervised approach rather than a fully supervised approach. Finally, there have been recent developments which use xGBoost to ensure generalization to different datasets."
> * reply : Thanks for your comment. scCA has a robust performance on the datasets we use in the experiments. But the suggestions you mentioned like using marker gene or xGBoost are helpful. In the future work we might consider making use of them.
>
> 2. * review : "The authors claim scCA provides greater interpretation compared to existing approaches. I do not believe the capsule features and compound features necessarily add to the biological interpretation of the cell-type gene expression states. The additional interpretation as claimed by the authors is not sufficiently elaborated."  and  "One of the main claims of the paper is that capsules encode similar information for cells of the same celltype. This claim is demonstrated by examples of few cells in Fig. 2. Given the centrality of the claim, the authors should elaborate and provide additional metrics to prove. The authors should also specify how the model architecture ensures this consistency."
> * reply : From the Visualization result, we discover that cells from different cell types have different patterns, including the Pearson Coefficients heatmap and attention weights heatmap. This helps to provide interpretations for scCA. scCA generates capsules to extract various features. By using LSTM and attention, scCA captures the relationships of these features and lay more emphasis on the important ones.
>
> 3. * review : "LSTM as noted by the authors is used for data with sequences. So, it is not clear why LSTM architecture is the choice here given there is no sequence data - in fact, the authors adjust the weights to account for the fact that single-cell data and capsules are not sequential in nature. Therefore, it is not clear why LSTM is the model choice compared to alternatives."
> * reply : Thanks for your question. We use a Bidirectional LSTM with attention as the main framework. The intention of LSTM and attention is that we hope to combine every feature’s information with the information of other features and lay more emphasis on those important ones. Furthermore, we are considering of an appropriate self-attention mechanism for single-cell RNA-seq cell type classification. The preliminary results are satisfying.
>
> 4. * review : "Given that single-cell data follows negative binomial distribution, variance is a function of the mean and hence use of most variable genes would not correct for the mean. Dispersion estimates are typically used as a work-around."
> * reply : Thanks for your comment. ZINB distribution is only a hypothesis distribution of gene expression level. Actually, there is not a consensus statistical hypothesis of single-cell RNA-seq data. In our article, we do not make use of the hypothesis that single-cell data follows ZINB distribution. We calculate the variances of genes and then pick out the top 2000 among them. Highly variable genes are more likely to help us distinguish different cells as they provide more information on cell heterogeneity. Also, there are tens of thousands of genes in human and mouse genome. Choosing the top 2000 largest variance genes also serves as dimension reduction of high dimensional scRNA-seq.
>
> 5. * review : "The gene expression map in Fig. 4 is rather unrepresentative since the dynamic range of scRNA-seq in linear scale spans 2-3 orders of magnitude"
> * reply : Thanks for your comment. Fig.4 is a gene expression matrix heatmap after data preprocessing. As Table1 in the paper shows, every dataset contains tens of thousands of genes. So here we offer 100 gene's expression level on 10 cells from dataset GSE72056 as a sketch map in the revised article. Also, because we employ a log transform in our data preprocessing, the range of expression level will not span 2-3 orders of magnitude.
>
> 6. * review : "Is the single-cell data normalized for differing molecule counts in each cell?"
> * reply : Yes, we normalize the expression levels into a reasonable range. For every cell in the data, we first divide every gene’s expression level by the sum of gene expression levels in the cell. This makes gene’s expression level among cells comparable. Then, we multiply the gene expression level with a constant 10^5 and employ a log transform to adjust the expression level in the data into a reasonable range. We have also revised and add detailed interpretations into the article.

---

### Decision · Program_Chairs · 2022-01-20

**Decision:**

Reject

**Comment:**

All reviewers believe that the paper is not ready for publication and clarity issue remain. All reviewers read the rebuttal responses, but they found that the paper wasn't revised during rebuttal, thus they retain their decisions.